# Effective prediction of biosynthetic pathway genes involved in bioactive polyphyllins in *Paris polyphylla*

Xin Hua [1,7], Wei Song[2,7], Kangzong Wang[1,7], Xue Yin[1], Changqi Hao[1], Baozhong Duan[3], Zhichao Xu [1,4✉], Tongbing Su [5,6✉] & Zheyong Xue [1✉]

The genes in polyphyllins pathway mixed with other steroid biosynthetic genes form an extremely complex biosynthetic network in *Paris polyphylla* with a giant genome. The lack of genomic data and tissue specificity causes the study of the biosynthetic pathway notably difficult. Here, we report an effective method for the prediction of key genes of polyphyllin biosynthesis. Full-length transcriptome from eight different organs via hybrid sequencing of next generation sequencingand third generation sequencing platforms annotated two 2,3-oxidosqualene cyclases (OSCs), 216 cytochrome P450s (CYPs), and 199 UDP glycosyltransferases (UGTs). Combining metabolic differences, gene-weighted co-expression network analysis, and phylogenetic trees, the candidate ranges of *OSC*, *CYP*, and *UGT* genes were further narrowed down to 2, 15, and 24, respectively. Beside the three previously characterized CYPs, we identified the OSC involved in the synthesis of cycloartenol and the UGT (PpUGT73CR1) at the C-3 position of diosgenin and pennogenin in *P. polyphylla*. This study provides an idea for the investigation of gene cluster deficiency biosynthesis pathways in medicinal plants.

[1] Key Laboratory of Saline-alkali Vegetation Ecology Restoration (Northeast Forestry University), Ministry of Education, Harbin, China. [2] College of Pharmacy, Zhejiang Chinese Medical University, Hangzhou, China. [3] College of Pharmaceutical Science, Dali University, Dali, China. [4] Institute of Medicinal Plant Development, Chinese Academy of Medical Sciences & Peking Union Medical College, Beijing, China. [5] Beijing Vegetable Research Center (BVRC), Beijing Academy of Agriculture and Forestry Science (BAAFS), Beijing, China. [6] National Engineering Research Center for Vegetables, Beijing 100097, China. [7]These authors contributed equally Xin Hua, Wei Song, Kangzong Wang. ✉email: zcxu@nefu.edu.cn; sutongbing@nercv.org; xuezhy@126.com

P. *polyphylla* var. *yunnanensis* is a member of Liliaceae family and one of the most famous medicinal plants in China. The rhizome of this plant is an important component of the traditional Chinese medicines "Yunnan Baiyao" and "Gongxue Ning"[1], which have pharmacological activities, such as hemostasis, analgesic, sedation, anti-inflammatory, and anti-tumor effects[2–4]. The main active components of this plant are steroidal saponins, also known as polyphyllins, accounting for about 80% of the total number of active compounds[5]. Polyphyllins have aroused great interest for their rich pharmacological activities, including anti-inflammatory, vascular protection, hypoglycemic, immunomodulatory, antiparasitic, hypocholesterolemic, antifungal, anti-parasitic, and anti-tumor effects[4,6,7]. However, the species is at risk of extinction due to its slow growth and excessive exploitation[8]. Given their complex molecular structures, polyphyllins are unlikely to be chemically synthesized for commercial usages. Therefore, metabolic engineering may be an effective method to provide a stable source of polyphyllins. The metabolic engineering strategy largely relies on the biosynthetic pathway of polyphyllins, which still has not been fully elucidated.

Polyphyllins are a group of products with different sugar chains connected at the C-3 or C-26 position of diosgenin or pennogenin. Diosgenin is also an important precursor to the synthesis of over 200 steroidal drugs (e.g., contraceptives, testosterone, progesterone, and glucocorticoids)[8]. However, the sources of diosgenin mainly depend on the extraction from several specific plants, such as yam (*Dioscorea* genus) and fenugreek (*Trigonella foenum-graecum*). The biosynthetic pathway of polyphyllins begins with the condensation of two molecules of isopentenyl diphosphate and one molecule of dimethylallyl diphosphate, which is then catalyzed by farnesyl diphosphate synthase (FPS) to form farnesyl diphosphate (FPP, C15)[9]. Two FPP molecules are catalyzed by squalene synthase (SQS) to produce a linear C30 molecule, squalene, which is further cycled by squalene epoxidase to 2,3-oxidosqualene[9]. Then, 2,3-oxidosqualene is cyclized by a cycloartenol synthase (CAS) to form cycloartenol, which is then modified through a series of oxidation and reduction to form cholesterol[10,11]. Enzymes in the CYP90G family can catalyze the hydroxylation of cholesterol C-16 and C-22 with the closure of E ring. Then, *16S,22S*-dihydroxycholesterol is further hydroxylated at C-26 and forms an F ring to produce diosgenin under the action of cytochrome P450s (CYPs), such as PpCYP94D108[12]. However, how steroidal skeleton α-hydroxylates at C-17 form pennogenin is still unknown. Subsequently, diosgenin and pennogenin are glycosylated by UDP glycosyltransferases (UGTs) to form various polyphyllins (Fig. 1). To date, the UGTs related to polyphyllin biosynthesis have still not been selected and functionally identified.

Although genomic and transcriptional information of numerous medicinal plants has been generated and made available to the public, the progress of candidate gene mining and whole pathway dissection of specialized plant metabolites remains slow due to the following factors[6,12–15]. The biosynthesis of several specified metabolites, such as the ginsenosides in Araliaceae family, lacks tissue specificity. A variety of ginsenosides are widely found in multiple tissue parts of the plant, and the complex distribution pattern of ginsenoside components hinders the prediction of the exact genes involved in biosynthetic pathway application by simple differential expression analysis[16]. The divergent evolution of CYP and UGT families generated numerous individual members that are phylogenetically close and can decorate diverse type of natural products. For this reason, predicting the related pathway gene solely based on phylogenetic analysis is impossible[17]. A previous study reported the extremely huge genome size of Parideae species (about 50 pg) (Pellicer et al., 2014). In addition, the biosynthetic genes of specific plant metabolites are scattered in different regions of the genome, further increasing the difficulty in identifying candidates precisely by the physical distance of metabolic pathway-related genes[18,19], despite the generation and good assembly of whole-genome sequences[20,21]. Therefore, an efficient strategy needs to be developed and improved urgently to accurately predict the key genes in the complex none-clustered biosynthetic pathway of specialized plant metabolites.

In this study, full-length transcriptome analysis using hybrid sequencing strategy based on single-molecule sequencing and paired-end mRNA sequencing was performed on eight different tissues from *P. polyphylla* var. *yunnanensis*. The weighted gene co-expression network analysis (WGCNA) combining the distributions of specific metabolites, different gene expressions, and phylogenetic analysis was further used to predict the key genes involved in the biosynthesis of polyphyllins. Then, candidate genes containing several assumed 2,3-oxidosqualene cyclase (OSC) genes and UGTs were functionally verified. This study may provide a strong basis for characterizing the steps of biosynthesis of polyphyllins of *P. polyphylla* var. *yunnanensis*, thus promoting the production of such important chemicals via synthetic biology.

## Results

**Transcriptome sequencing, assembly, and functional annotation.** A total of 292.69 Gb clean data for 21 sequencing libraries, including three biological replicates of rhizome, fibrous root, stem, leaf, and ripe fruit and two biological replicates of stigma, anther, and petals, were obtained by Illumina sequencing (Supplementary Table 1). A total of 81.81 Gb clean data containing 1,121,119 CCS reads were obtained from the PacBio sequencing platform. Among them, 969,450 long reads belong to the full-length non-chimeric sequence. The mean read length of CCS was 2263 bp. The full-length non-chimeric sequences were clustered into 69,009 consensus sequences, and the consensus sequences were polished using Quiver to obtain 68,266 high-quality consensus sequences. The low-quality consensus sequences were further corrected using the Illumina short reads. After removing redundant sequences for the high-quality consensus sequences and corrected low-quality consensus sequences, 39,875 transcript sequences were finally obtained. Using BUSCO[22] to evaluate the integrity of the transcriptome, the results showed that complete and single-copy duplicated transcript sequences accounted for 69.92%, the fragmented ones accounted for 5.97%, and those missing accounted for 26.11%.

First, we compared the known data in the nr database, annotating 38,177 (95.74%) unigenes from 39,875 transcripts. Afterward, we performed comparisons in the Swissprot, eggNOG, KOG, and Pfam databases and annotated 29,475 (73.92%), 37,673 (94.48%), 24,581 (61.65%), and 33,607 (84.28%) unigenes.

**Phylogenetic analysis of OSC, CYP, and UGT gene families.** In this study, we identified 2 intact OSCs, 216 CYPs, and 199 UGTs using PFAM annotation and BLAST algorithm. The phylogenetic tree of two OSCs from *P. polyphylla* var. *yunnanensis* and other 51 species showed that different branches distinguished the classification of OSC genes. The different OSC subfamilies are distributed in terms of various activities, especially for the skeletons of catalytic products, including cycloartenol, β-amyrin, lanosterol, lupeol, α-amyrin, friedelin, dammarenediol II, and mixed products (Fig. 2a). Two identified OSC transcripts from *P. polyphylla* var. *yunnanensis* were classified into the CAS clade.

Phylogenetic analysis of CYPs using *Arabidopsis* as a reference indicated that all the full-length P450s from *P. polyphylla* var. *yunnanensis* can be assigned to CYP51, CYP71, CYP710, CYP711, CYP72, CYP74, CYP85, CYP86, and CYP97 family. Among them, the CYP71 (85) and CYP86 (60) families have the

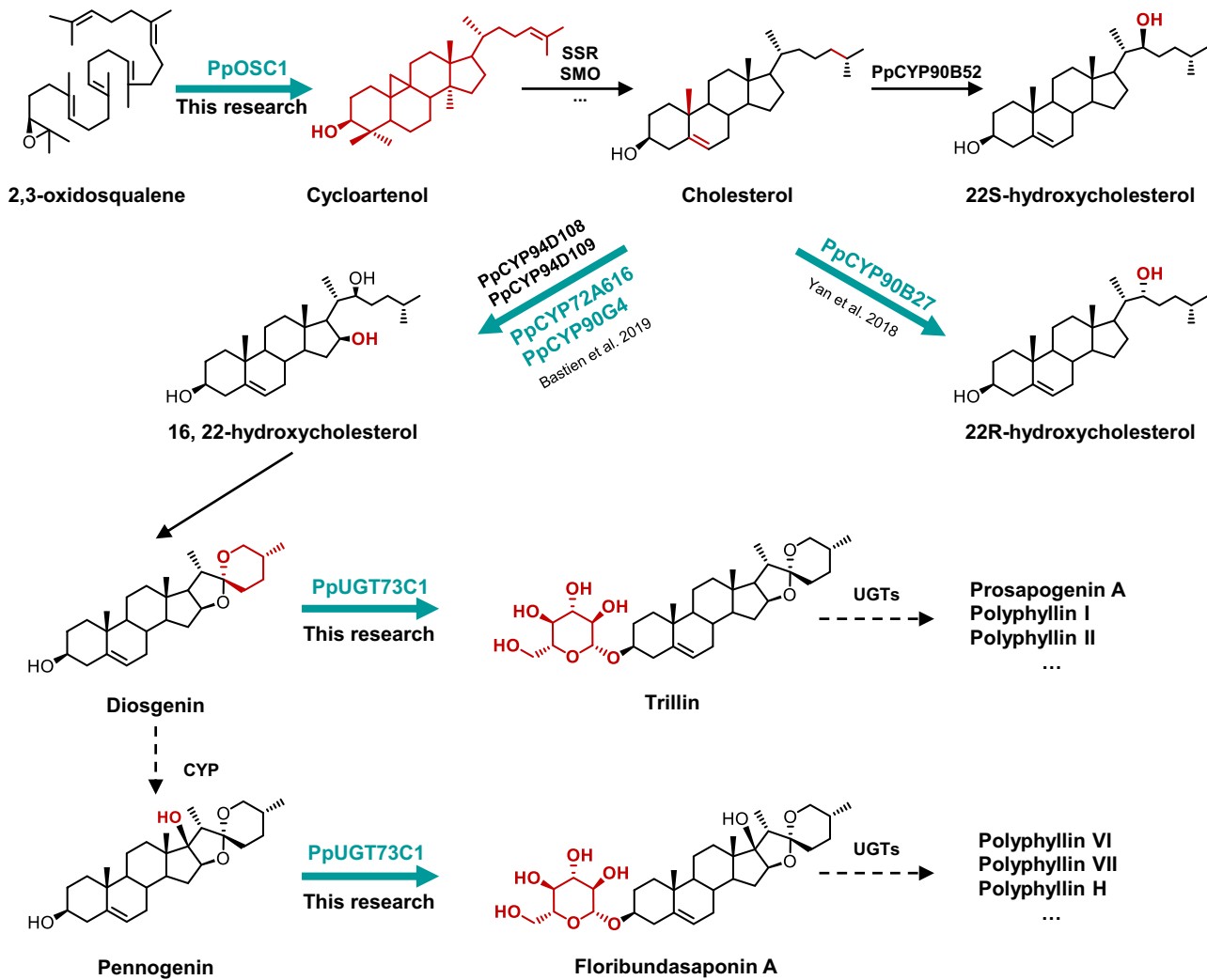

**Fig. 1 Possible biosynthetic pathways of polyphyllin in *P. polyphylla* var. *yunnanensis*.** The established metabolic pathways are represented by solid line arrows, while the speculated metabolic pathways are represented by dotted line arrows. Genes predicted by transcriptome, metabolite profile, and WGCAN analysis are shown in the yellow background, and those identified by this method are shown in the blue background.

largest number of P450 genes, followed by the CYP85 (37) and CYP72 (21) families, whereas CYP710 (4), CYP97(4), CYP74 (2), CYP711 (2) and CYP51 (1) has the smallest number of P450 genes (Fig. 3a and Supplementary Fig. 1).

The phylogenetic analysis of *P. polyphylla* var. *yunnanensis* UGTs was carried out together with the UGTs from *A. thaliana* and *Z. mays*, and the predicted UGT protein sequences of *P. polyphylla* var. *yunnanensis* were clustered into 13 of the 21 known UGT subfamilies[23]. The H, K, M, or N subfamilies of UGTs are lost in *P. polyphylla* var. *yunnanensis*. Among all the subfamilies, D is the largest phylogenetic group in *P. polyphylla* var. *yunnanensis*, containing 57 genes accounting for 28.64% of all UGTs (Fig. 3a and Supplementary Fig. 2).

**Metabolite and gene co-expression analysis to predict functional genes**. According to the results in Supplementary Fig. 3, a soft threshold $\beta = 9$ was selected to build a co-expression network. Then, the function hclust was used to perform hierarchical clustering on dissimilar matrices, whereas Dynamic Tree Cut was utilized to cut the generated cluster tree (Supplementary Fig. 4). In this process, unigenes with similar expression patterns can be combined on the same branch, and each branch represented a co-expression module, with different colors representing various

modules. Differential unigenes were correlated and clustered based on their FPKM values. Unigenes with a high correlation were assigned to the same module (Supplementary Fig. 5). In the end, 31,937 unigenes were divided into 26 modules, and the number of unigenes in the modules was 53–7667.

Using LC–MS–MS, we determined the production profiles of eight metabolites (cholesterol, diosgenin, trillin, prosapogenin A, polyphyllin I, polyphyllin II, polyphyllin VI, and polyphyllin VII) in different tissues of *P. polyphylla* var. *yunnanensis* (Fig. 3c and Supplementary Data 1). Diosgenin and most of its related metabolites (prosapogenin A, polyphyllin I, and polyphyllin II) had relatively similar distribution patterns and were highly accumulated in rhizomes, leaf, ovary, and petal. The distribution of trillin was exceptionally higher in leaf than in other tissues. The accumulation of pennogenin-derived saponins in different tissues of *P. polyphylla* also showed similar patterns. Polyphyllin VI was highly accumulated in rhizomes, fibril, fruit, and ovary, whereas polyphyllin VII was highly accumulated in fibril, fruit, ovary, and petal. The distribution patterns of diverse polyphyllins and transcriptome data were integrated to construct a co-expression network of metabolic pathway genes and metabolites (Supplementary Fig. 6). The network can combine gene modules and metabolites with similar patterns in various tissues of *P.*

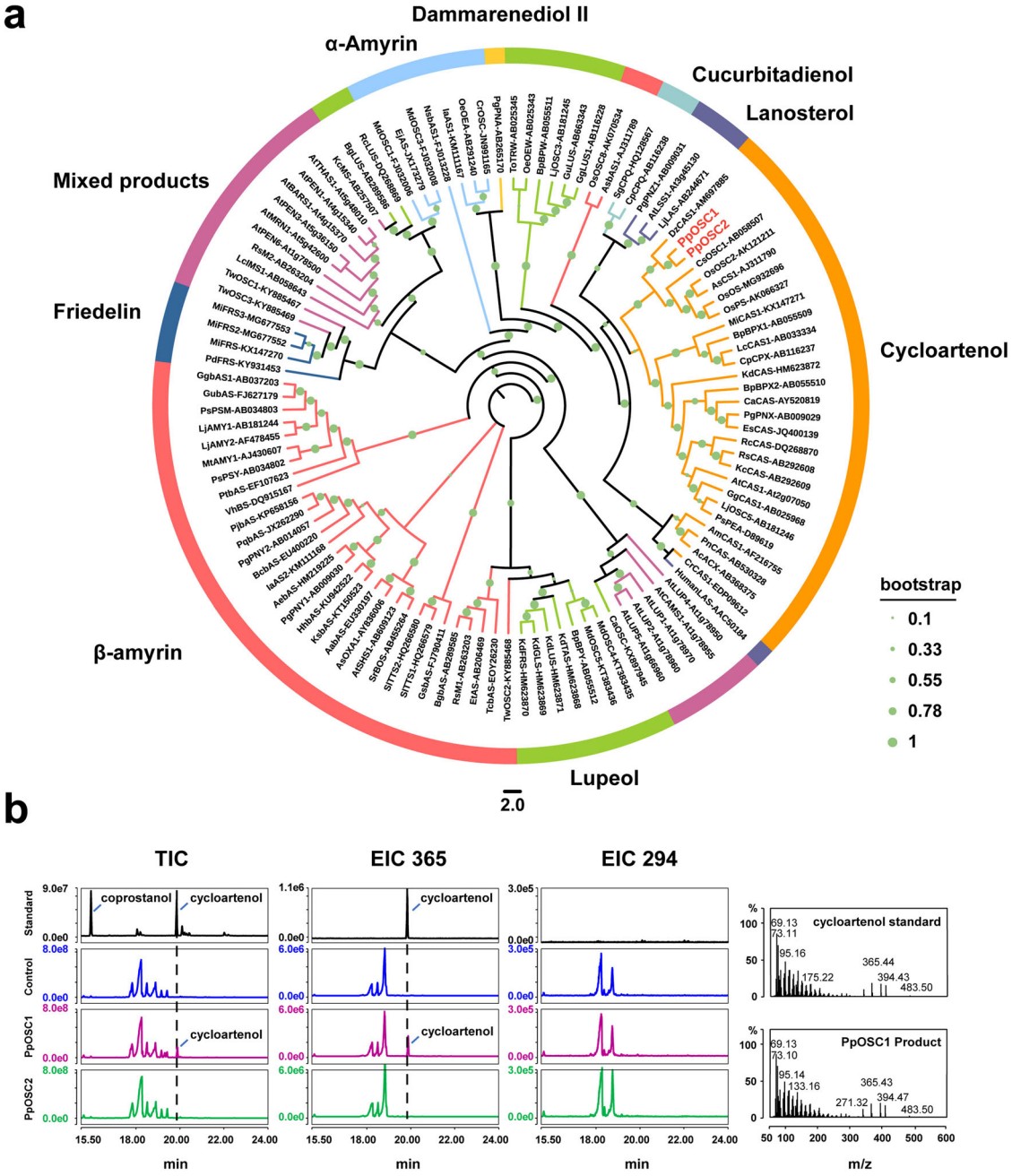

**Fig. 2 Candidate OSC genes and gene function verification. a** Phylogenetic tree of OSCs. Predicted amino acid sequences of OSCs in *P. polyphylla* var. *yunnanensis* were aligned with selected OSCs from other plant species using MUSCLE. The evolutionary history was inferred using the maximum-likelihood method. The bootstrap consensus tree inferred from 1000 replicates is taken to represent the evolutionary history of the taxa analyzed. **b** Functional verification of *PpOSC* gene. Two OSC genes were identified in *P. polyphylla* var. *yunnanensis*, and the results of GC-MS showed that *PpOSC1* gene increased the yield of cycloartenol after being transferred to *N. benthamiana*.

*polyphylla* var. *yunnanensis* and calculate the correlation coefficient between metabolites and gene modules (Fig. 3b). When the correlation coefficient approaches 1, the metabolites and genes in the module show similar expression or distribution patterns in different tissues of *P. polyphylla* var. *yunnanensis*.

Based on the correlation coefficient, we observed that trillin clustered well with the lavenderblush3 (0.82), lightblue3 (0.72), and white (0.73) modules in the co-expression network. Polyphyllin I had the highest correlation with Coral (0.72) and orangered4 (0.7) modules, and polyphyllin VII showed a high correlation with antiquewhite4 (0.81) and antiquewhite1 (0.76). Diosgenin and cholesterol were correlated with the antiquewhite4

module, and the correlation coefficients were 0.66 and 0.65, respectively. Polyphyllin II was correlated with lavenderblush3 (0.64) module, but polyphyllin VI and prosapogenin A showed no specific correlation with a certain module. The clustering results indicated that the genes in the orangered4, lavenderblush3, lightblue3, coral, antiquewhite1, white, and antiquewhite4 modules are likely to be involved in the biosynthesis of polyphyllins. Furthermore, we analyzed the upstream genes involved in the metabolic pathways of cholesterol or other phytosterols in the above seven modules. The coral, antiquewhite4, and lavenderblush3 modules contained relatively rich sterol synthesis upstream genes (14, 21, and 11, respectively)

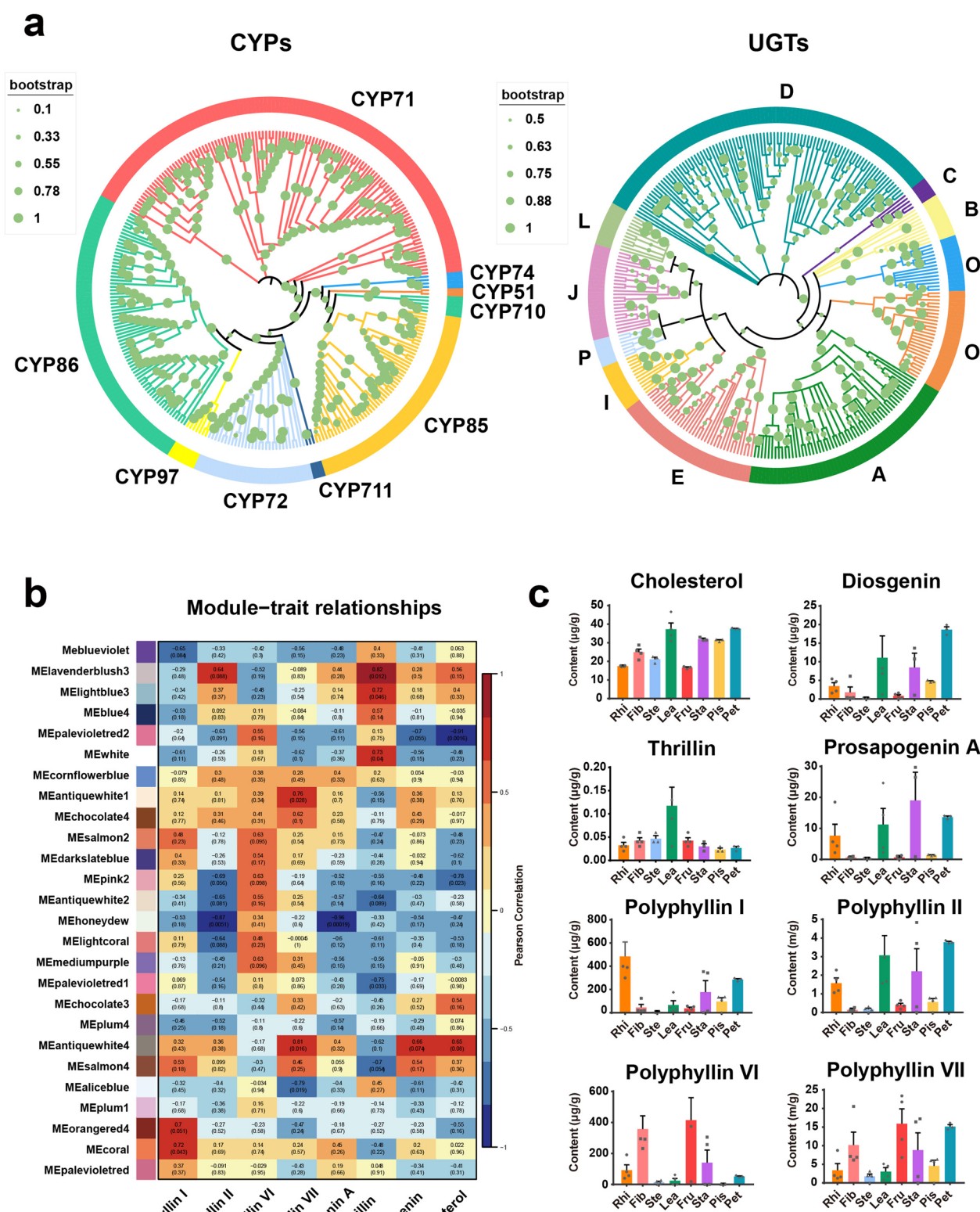

**Fig. 3 Using phylogenetic tree, metabolic profile, and WGCAN analysis to predict that the key genes involved in the biosynthesis of polyphyllin in *P. polyphylla* var. *yunnanensis*. a** Phylogenetic tree of CYPs and UGTs. **b** Module–trait associations. Each row corresponds to a module of characteristic genes, and each column corresponds to a metabolite. Each cell contains the correlation and *p*-value of the genes in the module with the corresponding metabolite. **c** The production profiles of key metabolites involved in the biosynthetic pathway of polyphyllin in different tissues. The quantification of Polyphyllins contents was carried out in four separate experiments, in which each sample came from different mixtures of four plants. The error line represents the standard deviation.

(Supplementary Table 2). However, the orange, lightblue3, white, and antiquewhite1 modules contained fewer sterol synthesis upstream genes (0, 0, 4, and 6, respectively). Based on the results, the genes contained in the coral, antiquewhite4, and lavender-blush3 modules are likely to play a key role in the biosynthesis of polyphyllins, and subsequent gene prediction will be developed around these three modules.

From the three predicted modules, we obtained 42 CYPs and 48 UGTs. Heat maps were plotted to show the expression patterns of these genes in different tissues (Fig. 4a). Genes involved in specialized metabolites biosynthesis often show high expression levels in certain tissues. Therefore, among the genes that have been predicted in our correlation analysis, the *CYP* and *UGT* genes with high expression levels in polyphyllins that accumulated tissues are likely to participate in the biosynthesis of polyphyllins.

In addition, we predicted the candidate *CYP* and *UGT* genes using phylogenetic analysis. P450s that can be hydroxylated in triterpene or steroidal skeletons often belong to the CYP72, CYP90, and CYP94 families. Based on our phylogenetic tree, we detected 21, 11, and 28 genes from the CYP72, CYP90, and CYP94 families, respectively. The UGTs that can add a glycosyl group at the C-3 position of triterpenoid and steroidal aglycone belong to the UGT73 family. A total of 57 UGT73s were annotated from our transcriptome data. Through combinational analysis of phylogenetic tree and WGCNA, we narrowed the *CYPs* and *UGTs* to 15 and 24 candidate genes, respectively (Fig. 4b). Among them, three CYP genes, namely, *PpCYP90G4* (F01_transcript/40556), *PpCYP90B27* (F01_transcript/40246), and *PpCYP72A616* (F01_transcript/40246), have been reported to be involved in the biosynthesis of polyphyllins. These three genes ranked 1st, 3rd, and 6th place in the list of candidate genes, respectively. These candidate genes are very likely to be involved in the biosynthesis of polyphyllins. The genes that appeared in the candidate clades of phylogenetic tree but were not included in the candidate modules of WGCNA should not be ignored. They may also be involved in the biosynthesis of polyphyllins. All candidate *OSC*, *CYP*, and *UGT* genes can be found in GenBank (BankIt2522666: OL654188–OL654276).

### PpOSC1 but not PpOSC2 catalyzes the conversion of 2,3-oxidesqualene to cyclic triterpenes.
OSC is one of the key enzymes in steroid biosynthesis, and the OSC-catalyzed conversion of 2,3-oxidesqualene to cyclic triterpenes marks the first scaffold diversification reaction in triterpenoid and steroid pathways. In contrast to single-copy *OSC* gene in lower plants, higher plants always have multiple *OSC* gens in their genomes[9]. A total of 13 OSC-related transcriptome variants were found in the transcriptome, which may represent two non-redundant *OSCs* (*PpOSC1* and *PpOSC2*). To investigate the activity of the two candidates, we cloned two genes transferred them into optimized yeast strains (SQSQ-pPOSC1 and SQSQ-pPOSC2). Cycloartenol production was evidently observed in the SQ-PPOSC1 strains, whereas it was absent in SQ-PPOSC2 strains, suggesting that *PpOSC1* gene encodes CAS in *P. polyphylla* var. *yunnanensis* (Fig. 2b). Furthermore, *PpOSC1* and *PpOSC2* genes were transfected into *A. tumefaciens* for infected *N. benthamiana* leaf infiltration. Similarly, after infiltration, we collected the leaves and verified the function of *OSC* genes by measuring the content of triterpenes. The results showed that the instantaneous expression of *PpOSC1* can facilitate the production with 3.12-fold more cycloartenol, whereas *PpOSC2* could not increase the content of cycloartenol but can increase that of an uncharacterized triterpene product instead (Supplementary Fig. 7). Therefore, the *PpOSC1* gene plays an important function in the biosynthesis of polyphyllin.

### PpUGT73CR1 functions as a glucotransferase at the C-3 position of diosgenin and pennogenin.
We verified the glycosyltransferase activity of candidate UGT genes (marked with asterisk in Fig. 4a) at C-3 position of diosgenin, and the results showed that only transcript/33044 showed activity and was named as *PpUGT73CR1* (Supplementary Figs. 8 and 9). In the phylogenetic tree, *PpUGT73CR1* is in the same branch with *BvUGT73CR10* and *BvUGT73CR10*, which can glycosylate the C-3 position of β-amyrin[24]. The coding sequence length of the gene is 1473 bp, encoding a UGT with 490 amino acid residues. The molecular weight of the protein is 54.48 kDa, and the molecular weight of the fusion protein with GST tag is 81.68 kDa (Fig. 4c and Supplementary Fig. 10). When diosgenin was used as a sugar acceptor, the enzyme encoded by *PpUGT73CR1* can catalyze diosgenin to produce a polar product after the addition of UDP-glucose. The HPLC retention time of the product was 22.1 min, which was consistent with that of trillin. When pennogenin was used as the sugar acceptor, PpUGT73CR1 converted pennogenin into a new product with a retention time of 18.9 min. The LC-TOF-MS analysis showed that the molecular weight of the new product was 593.37 $[M + H]^+$, which was consistent with the molecular weight of floribundasaponin A (Fig. 4d). Floribundasaponin A is a product of glycosylation of the C-3 position of pennogenin and has been found in *Dioscorjgi Floribunda* and *Paris Polyphylla* var. *chinensis*[25,26]. Supplementary Note 1 shows the hydrogen and carbon spectrum results of the substrate (pennogenin) and reaction product (floribundasaponin A).

To study the promiscuity of UGT to diverse substrates, we evaluated the catalytic capability of PpUGT73CR1 on diosgenin and pennogenin. Subsequently, the enzymatic kinetics of PpUGT73CR1 catalyzing different substrates were studied (Table 1). The maximum reaction velocity ($V_{max}$) of diosgenin and pennogenin were $177.13 \pm 8.91$ and $87.7 \pm 3.27$ nmol/min/mg, respectively. Michaelis constant ($K_m$) reflected the affinity between the enzyme and substrate to a certain extent. Compared with pennogenin ($K_m = 73.43 \pm 8.16$ μM), PpUGT73CR1 had a higher affinity for diosgenin ($K_m = 53.69 \pm 9.37$ μM). The enzymatic catalytic constant ($K_{cat}$) of PpUGT73CR1 for diosgenin and pennogenin were 0.24 and 0.12 $s^{-1}$, respectively, and the calculated conversion efficiencies ($K_{cat}/K_m$) were 4.47 (diosgenin) and 1.62 (pennogenin) $mM^{-1} s^{-1}$. Thus, the catalytic efficiency of PpUGT73CR1 for diosgenin was higher than that of pennogenin.

### Discussion
Medicinal plants are rich in a variety of metabolites with pharmacological properties. However, studies on the biosynthetic pathways of these metabolites are still limited due to the complexity of plant genomes and the lack of genomic resources. Unlike microorganisms, functional gene clusters are rare in plants, and gene redundancy and strict genetic regulation in plants cause difficulty in parsing metabolic pathways[29]. At present, botanists mainly use multi-omics methods to analyze the biological pathways of metabolites. However, in the absence of genomic data, this prediction method often yields a large number of false-positive results. For the majority of medicinal plants, an accurate method is needed to predict metabolite biosynthesis pathways without genomic data or metabolic biosynthesis clusters. The research on the biosynthesis of polyphyllin has been a hot topic because of its various pharmacological activities. Given the huge genome, no data report is currently available on the genome of *P. polyphylla* var. *yunnanensis*. At this stage, transcriptome sequencing is the most suitable method to study the biosynthesis pathway of polyphyllin.

Several studies reported the transcriptome data of *P. polyphylla*[6,14,15,30]. However, these transcriptome measurements

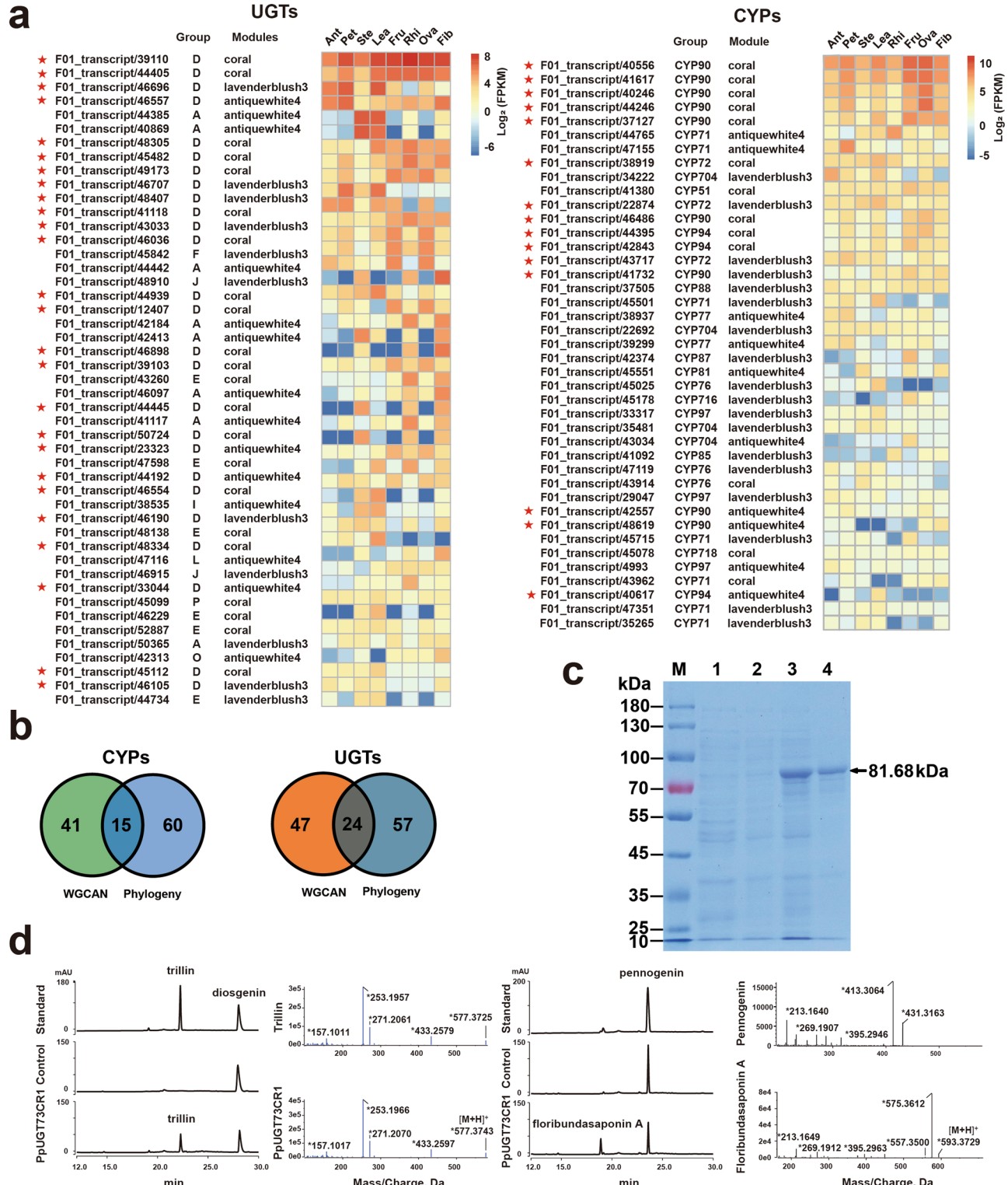

**Fig. 4 Candidate CYP and UGT genes and gene function verification. a** Heatmaps of the expression levels of candidate CYPs and UGTs in different tissues of *P. polyphylla* var. *yunnanensis*. All genes are arranged from top to bottom according to the total expression level. The asterisks represent key genes predicted by the evolutionary tree, WGCAN and gene expression. **b** Venn diagrams of candidate genes. Phylogenetic tree and WGCAN methods were used to predict candidate CYPs and UGTs, among which 15 and 20 CYPs and UGTs could be predicted by the two methods, respectively. **c** SDS–PAGE analysis of expressed PpUGT73CR1 protein. *Lanes*: M, protein molecular weight marker (Thermo Fisher); 1, pGEX-6p-1 vector transformed in *E. coli* Rosetta (DE3) cells with IPTG induction; 2, pGEX-*UGT73CR1* vector transformed in *E. coli* Rosetta (DE3) cells without IPTG induction; 3, pGEX-*UGT73CR1* vector transformed in *E. coli* Rosetta (DE3) cells with IPTG induction; 4, the purified recombinant protein of PpUGT73CR1. **d** Functional verification of *PpUGT73CR1* gene. The functional verification of candidate UGTs was performed by HPLC and LC-TOF-MS, and the enzyme encoded by *PpUGT73CR1* gene could introduce the glucose group into C-3 of diosgenin and pennogenin.

**Table 1 The enzymatic kinetics of PpUGT73C1 catalyzed diosgenin and pennogenin.**

| Enzyme | Substrate | $V_{max}$ (μM/min) | $K_m$ (μM) | $K_{cat}$ ($s^{-1}$) | $K_{cat}/K_m$ ($s^{-1}$ $mM^{-1}$) |
|---|---|---|---|---|---|
| PpUGT73C1 | Diosgenin | 1.771 ± 0.089 | 53.69 ± 9.37 | 0.24 | 4.47 |
| | Pennogenin | 0.877 ± 0.033 | 73.43 ± 8.16 | 0.12 | 1.62 |

were all based on Illumina platform and cannot reflect well the complete transcriptome information of *P. polyphylla* var. *yunnanensis*. Published transcriptome information was mainly derived from roots, stems, and leaves of the plant, with little transcriptome information for other tissues, which is insufficient to predict polyphyllin biosynthesis pathway genes using the association analysis of gene expression and metabolites. In this study, we collected samples from eight tissues of *P. polyphylla* var. *yunnanensis* to complete the splicing and full-length transcriptome sequencing, and more than 370 G clean data were obtained. The transcriptome data in our study obtained the most diversity in tissues and deepest sequencing depth among all transcriptome experiments in *P. polyphylla* var. *yunnanensis*. Compared with previous reports of sequencing, it can avoid redundant data and splicing errors and provide data support for the accurate prediction of the biosynthetic pathway of polyphyllin.

We constructed a co-expression network of metabolites with gene expression levels in different tissue of *P. polyphylla* var. *yunnanensis*. We measured the contents of key metabolites related to polyphyllin synthesis in various tissues of *P. polyphylla* var. *yunnanensis*, including cholesterol, diosgenin, trillin, prosapogenin A, polyphyllins I, II, VI, and VII. We predicted three modules that are highly relevant to the above metabolites. We further conducted conditional screening through phylogenetic trees and gene expression levels and obtained reliable candidate genes, including three reported key *CYP* genes involved in the biosynthesis of polyphyllins. We also identified an *OSC* gene responsible for cycloartenol biosynthesis and a *UGT* gene with C-3 glucosyltransferase function from *P. polyphylla* var. *yunnanensis*. Among the predicted modules, the coral module showed a strong correlation only with polyphyllin I, whereas the lavenderblush3 and antiquewhite4 modules exhibited more correlations with cholesterol, diosgenin, and trillin. This finding suggests that the genes contained in the coral module may be more involved in the formation of hydroxylation and glycosylation of polyphyllins. The predicted results also proved our speculation that the three predicted CYP genes with a clear function and the C-3 glucosyltransferase gene of polyphyllins (*PpUGT73CR1*) all come from the coral module. These results showed the accuracy and reliability of this prediction method.

Several studies focused on glycosylation modification of steroidal sapogenins. A C-3 glycosyltransferase SAGT4a in *Solanum aculeatissimum* shows the glycosylation activity of diosgenin, nuatigenin, tigogenin, and other glycosyltransferases[31]. In this study, c-3 glycosyltransferases of diosgenin and pennogenin were found in *P. polyphylla* var. *yunnanensis*. According to the study of substrate promiscuity and enzyme kinetics, PpUGT73CR1 had a better affinity and catalytic capability with diosgenin compared with pennogenin.

At present, the analysis of steroidal sapogenin biosynthetic pathways is progressing slowly and remains in the exploratory stage. Research has focused on the cloning and regulation of functional genes upstream of terpenoid biosynthetic pathways, such as HMGR, FPS, SS, CAS, etc.[32,33], and several P450s. Other research reported the glycosylation modification of diosgenin. Therefore, the key genes involved in the biosynthesis pathway of polyphyllin can be possibly predicted by using our prediction method combining the evolutionary tree, co-expression network, and gene expression quantity. This method is also generally applicable to the prediction of key genes in plants lacking genome data.

Polyphyllin has a variety of pharmacological activities, but the analysis of the biosynthetic pathway of polyphyllin is incomplete. We performed splicing and full-length transcriptome sequencing of rhizomes, fibrous roots, stems, leaves, ripe fruits, stigma, petals, and pistil tissues of *Paris polyphylla* var. *yunnanensis*, and the gene expression and WGCNA method were used to predict the OSCs, CYPs, and UGTs involved in the biosynthesis of sapogenin. Among the predicted candidate genes, we identified an *OSC* gene (*PpOSC1*) and a diosgenin/pennogenin C-3 UGT gene (*PpUGT73CR1*). This study improves our understanding of the biosynthetic pathways of polyphyllins, providing a basis for further elucidation of the pharmacologically active triterpene/sapogenin biosynthesis and an efficient strategy to study the complex pathway of other specialized plant metabolites.

## Methods

**Plant material and RNA preparation.** Samples of 4-year dwarf *P. polyphylla* var. *yunnanensis* were collected from Dali, Yunnan, China. The rhizomes, fibrous roots, stems, leaves, ripe fruits, stigma, petals, and pistil were harvested in 5-year-old healthy plants (Supplementary Fig. 11). The rhizomes, fibrous roots, stems, leaves, and ripe fruits were harvested in October 2018, whereas the stigma, petals, and pistil were harvested in April 2019. All tissues were frozen in liquid nitrogen immediately and stored at −80 °C after collection. Every sample had three biological replicates that were sequenced independently.

**RNA isolation, transcriptome sequencing, and gene function annotation.** The sequencing samples of rhizomes, fibrous roots, stems, leaves, ripe fruits, stigma, petals, and pistil were from multiple plants, and the full-length transcriptome sequencing samples were from a mixture of different tissues from multiple plants. Total RNA was isolated using an RNA Plus kit (Takara, Qingdao, China), in accordance with the manufacturer's protocol. Three biological replicates of rhizome, fibrous root, stem, leaf, and ripe fruit were determined, and two biological replicates of stigma, anther, and petal were tested due to difficulty in sample collection and insufficient RNA quality. RNA quality was examined using Agilent 2100 (Agilent Technologies, Santa Clara, USA). The cDNA library was constructed and sequenced by Biomarker Technologies Corporation (Beijing, China). A single tissue was sequenced using an Illumina Hiseq 2000 platform, and full-length transcriptome sequencing for the mixture of different tissues was performed using PacBio Sequel platform. The PacBio long reads were filtered, and redundant sequences were removed using CD-HIT-EST program. In full-length transcriptome data, sequences with polymerase read <50 bp and sequence accuracy <0.9 were filtered out. Then, the clean reads from Illumina sequencing were mapped into the non-redundant long-reads to calculate the fragments per kilobase per million (FPKM) values using DESeq2. Specifically, FPKM = cDNA fragments/mapped fragments (millions) × transcript length (kb), where cDNA fragments represent the number of fragments aligned to a transcript, mapped fragments (millions) is the total number of fragments aligned to the transcript; transcript length (kb) denotes the transcript length. The filtered full-length transcripts were functionally annotated using non-redundant (nr), SWISSPROT, Gene Ontology, Clusters of Orthologous Genes, EuKaryotic Orthologous Groups (KOG), PFAM, and Kyoto Encyclopedia of Genes and Genomes databases, respectively.

**Phylogenetic analysis.** Transcripts belonging to OSCs, CYPs, and UGTs were identified using BLAST software. The transcripts with a length under 1000 bp were removed. Supplementary Data 2 lists the OSC sequences from different plants used to construct the phylogenetic tree. The phylogenetic trees of OSCs, P450s, and UGTs were constructed using the maximum likelihood method and Jones–Taylor–Thornton model using MEGA 7.0[27]. A bootstrap resampling analysis with 1000 replicates was performed to evaluate the topology of phylogeny.

**Metabolite content determination.** The tissues frozen at −80 °C were lyophilized in a freeze dryer. Then, 20 mg dry materials from different tissues were weighted and placed in a 2 mL centrifuge tube. A total of 1 mL 80% methanol containing 20 μg internal standard was added to the sample (digitoxin, ≥95%, Sigma). The

samples were further extracted at 1400 rpm for 2 h and centrifuged at $10,000 \times g$ for 5 min. The supernatant was transferred to a new centrifuge tube and added with 300 μL n-hexane for extraction. After extraction, the n-hexane layer was sucked and removed, and the process above was repeated. SpinVac was applied for solvent removal in the samples. The samples were redissolved with 500 ml distilled water. Then, extraction was performed twice with 500 ml n-butanol. Nitrogen was used to dry the organic phase of the sample and redissolved sample with the mobile phase during measurement.

The analysis was performed on a Waters ACQUITY ultra-performance liquid chromatography (LC) system coupled with an AB Sciex 5500 Qtrap mass spectrometer (AB Sciex, Milford, MA, USA). Chromatographic separation was achieved on an ACQUITY BEH C18 column ($100 \times 2.1$ mm$^2$, 1.7 μm) at 40 °C. The 0.1% formic acid water was used as mobile phase A, and 0.1% formic acid in acetonitrile was used as mobile phase B. The gradient was 0–1 min, 5–52% B; 1–6 min, 52–56% B; 6–7 min, 56–95% B; 7–7.5 min, 95–95% B; 7.5–9 min, 95–5% B; 9–10 min, 5–5% B. The flow rate was 0.25 mL min$^{-1}$, and the injection volume was 5 μL. The electrospray ionization source interface operated in negative ionization mode was used in this study. The ion spray voltage was set at −4500 V. Supplementary Table 3 shows the optimized multiple reaction monitoring parameters for the analytes and internal standard. Cholesterol detection was completed on a Thermo ISQ-LT gas chromatography–mass spectrometry (GC–MS) system using Thermo TG-5HT column (30 m × 0.25 mm × 0.10 μm). The mass detector was set to SCAN mode, the scanning range was 60–800 $m/z$, and the solvent delay was 10 min. The GC conditions were as follows. The sample (1 μL) was injected in split mode (10:1) at 250 °C under a He flow rate of 1.2 mL min$^{-1}$, and the temperature cycle involved the initial injection temperature of 170 °C for 2 min, 170–290 °C for 6 °C per minute, holding for 4 min after the temperature reached 290 °C, and raising the temperature from 290 to 340 °C at 25 °C per minute.

**Construction of gene co-expression networks**. Gene co-expression networks were constructed using the WGCNA approach with R packages (version 3.2.2)[28]. Here, we used the normalized quantile function in the R software package to normalize the gene expression data. We selected the expression matrix of 31,937 genes with the sum FPKM value in all tissues >1.0 from all genes as the input file for WGCNA to identify gene modules with strong co-expression. Before the construction of the network module, outlier samples should be removed to ensure the accuracy of the results because the analysis results of the network module are easily affected by outlier samples. By calculating the correlation coefficient of each sample's expression level and clustering, the samples with low correlation or those that cannot be clustered on the tree graph are removed. Next, WGCNA network construction and module detection were conducted using an unsigned type of topological overlap matrix (TOM). Based on the TOM, we used the average-linkage hierarchical clustering method to cluster genes, following the standard of the hybrid dynamic shearing tree and set the minimum number of genes for each gene network module to 30. The power $\beta$ was selected based on the scale-free topology criterion. The modules were detected as branches of the dendrogram using the dynamic tree-cut, and a cut-off height of 0.25 was used to merge the branches to the final modules.

Finally, the gene visual network was described by using a heatmap. The heatmap depicts the TOM among all genes in the analysis. Light color represents a low overlap, and the progressively darker red color represents higher overlap. Blocks of darker colors along the diagonal are the modules, and a very strong association existed between the genes that are contained within these red modules. These red modules were the focus of our genetic prediction.

**Functional verification of OSC genes**. The function of OSC gene was verified by yeast strain and *Nicotiana benthamiana*. Supplementary Table 4 shows all the selected strains and plasmids used in the yeast experiment. *PpOSC1* and *PpOSC2* were cloned from *P. polyphylla* var. *yunnanensis* and transferred into pδHis plasmid. The plasmid was transformed into yeast strain BY-SQ1 using the standard lithium acetate approach. The yeast strains SQ-PpOSC1 and SQ-PpOSC2 were precultured in 5 mL synthetic defined medium with glucose as carbon source and uracil and histidine omitted (SD-URA-HIS) at 30 °C and 220 rpm for 24 h. Precultures were inoculated at an initial optical density (OD)$_{600}$ of 0.05 in 50 mL SD-URA-HIS in 250 mL flasks and grown under the same condition for 72 h. The cells were harvested, resuspended in 2 mL 10% KOH (w/v) and 90% ethanol (v/v), heated for 2 h at 75 °C, and cooled and extracted once with 0.5 mL ethyl acetate. After centrifugation, the ethyl acetate phase was collected and dried by centrifugal vacuum evaporator. Derivatization of the dried products was conducted with 1-(trimethylsilyl)imidazole-pyridine mixture at 70 °C for 30 min to prepare the sample for analysis.

In the transient expression system of *Nicotiana benthamiana*, the coding regions of candidate OSC genes were cloned from *P. polyphylla* var. *yunnanensis* into the pEAQ-HT-DEST1 vector. After sequence verification, pEAQ-HT-DEST1 vectors carrying OSC genes were separately transferred into *Agrobacterium tumefaciens* strain GV3101 and cultured overnight at 28 °C and 220 rpm. Then, 1 mL culture was used to inoculate 10 mL Luria–Bertani (LB) medium containing 50 μg/mL kanamycin, 25 μg/mL rifampicin, and 25 μg/mL gentamicin for overnight growth. The following day, the cultures were centrifuged (5000×g, 5 min), and cells were resuspended in infiltration buffer (10 mM MES ($C_6H_{13}NO_4S$), pH 5.6, 10 mM MgCl$_2$, and 100 μM acetosyringone) to a final OD$_{600}$

of 0.4. The leaves of 6-week-old *N. benthamiana* were infiltrated with *A. tumefaciens* solution as follows. A 5 mL needle-free syringe was used to gently push the bacterial mixture into the abaxial surface until the entire leaf was filled with agrobacterium. The infiltrated leaves were cultured at 22 °C, exposed to light for 10 h a day, and harvested at 6th day after infiltration. For metabolite extraction, leaf disks in diameter 1 cm were prepared from *Agrobacterium*-infiltrated *N. benthamiana* and dried with a vacuum freeze-dryer. Then, the leaves were ground into powder, and 10 mg powder was weighed and placed a 2 mL tube for use. Then, 2 mL lysate was added to each sample and heated in the water bath (75 °C for 1 h). After the samples were completely dried, 300 μL ethyl acetate and 500 μL water were added and mixed with vortex shock and centrifuged for 10 min to facilitate separation. Next, 100 μL was removed from the upper layer (ethyl acetate layer) and transferred to a special glass tube. The liquid was blow-dried with nitrogen and added with 50 μL 1-(trimethylsilyl) imidazole-pyridine mixture. After vortex-mixing–heating at 70 °C for 30 min, the mixture was analyzed with GC–MS same as cholesterol analysis above.

**Cloning and prokaryotic expression of *UGT* genes from *P. polyphylla* var. *yunnanensis***. The total RNA from the *P. polyphylla* var. *yunnanensis* was extracted and reverse transcribed to obtain cDNA. The candidate polymerase chain reaction (PCR) primers for UGTs were designed based on the transcriptome sequence. The PCR procedure was as follows: 95 °C for 3 min; 95 °C for 30 s, 60 °C for 30 s, and 72 °C for 90 s in 33 cycles; 72 °C for 5 min. The primers of UGT genes are shown in Supplementary Data 3. The prokaryotic expression vector pGEX-6p-1 was linearized with restriction endonucleases EcoR I and Sal I (Thermo), recombined with the PCR product through the ClonExpress II One Step Cloning Kit (Vazyme), and transformed into *E. coli* DH5α.

The plasmid with correct sequencing was transformed into Rosetta-gami B (DE3) pLysS and inoculated into LB liquid medium containing ampicillin (100 mg/L) and then cultured at 37 °C at 180 rpm until the OD$_{600}$ of = 0.6. A total of 0.2 mM isopropyl β-D-1-thiogalactopyranoside was added to the culture medium, induced at 16 °C for 16 h, and centrifuged at 4 °C at 5000 rpm to collect the bacteria. The bacterial cells were suspended in 10 mM phosphate buffer (pH 7.4), and the cells were disrupted by ultrasound in an ice bath. Then, the cells were centrifuged at $12,000 \times g$ at 4 °C for 20 min. The bacterial supernatant was purified using glutathione beads (Smart-Life Sciences, Changzhou, China) and concentrated using Millipore ultrafiltration tubes (Meck, Darmstadt, Germany). Pierce BCA Protein Assay Kit (Thermo, Waltham, USA) was used to quantify the target protein.

*Enzyme activity analysis*. The enzymatic reaction system consisted of 50 mM Tris (pH 8.0), 1 mM MgCl$_2$, 5 mM glucose donor (UDP-glucose), 1 mM glucose receptor (diosgenin/pennogenin), and purified enzyme of PpUGT73CR1 in a final volume of 100 μL. After overnight incubation at 37 °C, an equal volume of ice methanol was added to stop the reaction. The product was concentrated and dried, dissolved in 100 μL chromatographic methanol, and centrifuged at 12,000 rpm for 10 min, and the supernatant was obtained for testing. The reaction products were identified by high-performance LC (HPLC) and LC time-of-flight mass spectrometry (LC-TOF-MS), and the Thermo Hypersil GOLD C18 column (250 mm × 4.6 mm, 5 μm) was used for HPLC detection. The mobile phases were water (A) and acetonitrile (B). The elution gradient was as follows: 0–6 min, 20–30% B; 6–15 min, 30–60% B; 15–21 min, 60–100% B; 21–30 min, 100% B; 30–35 min, 100–20% B. The flow rate was 1 mL/min, the column temperature was 30 °C, the injection volume was 10 μL, and the detection wavelength was 210 nm. LC-TOF-MS was performed using the AB Sciex Tripletof 6600 (AB Sciex, Milford, MA, USA) in a positive ionization mode.

For the kinetic analysis of UGT73CR1, the reaction mixture contained 50 mM Tris−HCl (pH 8.0), 5 mM UDP-glucose, acceptor substrate (20–400 μM diosgenin and pennogenin), and 1 μg purified UGT73CR1 in a final volume of 100 μL. The reaction was incubated at 37 °C for 30 min. HPLC analysis was used to quantify the target product in each reaction. The Michaelis–Menten parameters were calculated by kinetic model using Prism 7 (GraphPad, San Diego, CA, USA). All data are presented as means ± standard deviation of three independent experiments.

**Statistics and reproducibility**. All data calculations in this research are from more than three independent experiments. The WGCNA analysis and heatmap drawing of CYPs and UGTs were generated with R packages (version 3.2.2) and the concentration data of polyphyllins were processed by Graphpad 7.0 and presented with mean ± standard deviation.

**Reporting summary**. Further information on research design is available in the Nature Research Reporting Summary linked to this article.

## Data availability

Source data underlying Fig. 3c are presented in Supplementary Data 1. Raw reads have been deposited as a BioProject under accession PRJCA004404. The raw reads have been deposited in The National Genomics Data Center (NGDC), part of the China National Center for Bioinformation (CNCB). The accession number is PRJCA004404. The gene sequences of all the candidate genes presented in the text have been uploaded to GenBank, the accession numbers are BankIt2522666: OL654188–OL654276.

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

## Acknowledgements

This work was supported by the program for Natural Science Foundation of Heilongjiang Province of China (YQ2020C020), Opening Project of Zhejiang Provincial Preponderant and Characteristic Subject of Key University (Traditional Chinese Pharmacology), the Zhejiang Chinese Medical University (No. ZYAOX2018012), the National Natural Science Foundation of China (NSFC Grant No. 31770332, 31970314), National Key R&D Program of China (2019YFC1711103), and the Fundamental Research Funds for the Central Universities (2572020BD01) and Key project at central government level: The ability establishment of sustainable use for valuable Chinese medicine resources (2060302-2101-17). The authors also acknowledge the technical support of Dr. Shengnan Tan from Analysis and Test Center, Northeast Forestry University.

## Author contributions

Z.X. and X.H. designed the experiments and coordinated the project. K.W., X.Y. and C.H. performed the samples collection, phylogenetic tree, OSC function, transcriptomic. and metabolomic analyses. X.H. wrote and edited most of the manuscript. B.D. edited the language. All authors have read and approved the final manuscript.

## Competing interests

The authors declare no competing interests.
