## [Transparent Peer Review File · Communications Biology]

Reviewers' comments:

Reviewer #1 (Remarks to the Author):

In this manuscript, the authors reported an effective method for predicting key genes of polyphyllins biosynthesis, and claimed their study provides a new idea for the study of gene cluster deficiency biosynthesis pathways in medicinal plants. But after reading this manuscript carefully, I find it difficult to evaluate the reliability of their workflow and the accuracy of the results, because a lot of the details of the analysis and results are not provided. For example, the software parameters and thresholds used in the analysis process (such as how to calculate the fpkm values using `deseq2`), and whether the important functional genes (OSC, CYP, and UGT) obtained by them have been submitted to GenBank, and where to find their sequences. I think only by providing these details can we continue to discuss the correctness of the manuscript, otherwise we can only reject it.

Reviewer #2 (Remarks to the Author):

Many medicine plants have extremely huge and complex genome. In this manuscript, authors constructed an co-expression network of metabolites with gene expression levels using multi-tissue transcriptome sequencing, and successfully predicted the candidate OSCs, CYPs and UGTs involving in the biosynthesis of sapogenin. The PpOSC1 and PpUGT73C1 in polyphyllin biosynthetic pathway were further identified. The methods and results are logical and accurate for publication.

A few suggestions:

- 1) it is better to modify the title of this manuscript, and exclude "none-clustered metabolic pathway".
- 2) Double check the manuscript, since there are a lot of mistakes. For examples, there is a mistake in the Figure 4d, it should be PpUGT73C1, instead of PpUGT73E5? In addition, should it be clarified how PpUGT73C1 was selected for enzymic assay?

Reviewer #3 (Remarks to the Author):

Polyphyllins are a group of medicinal compounds with various pharmacological activities. They are produced in the plant *Paris polyphylla*, which is a medicinal herb widely used in Chinese traditional medicine. Structurally speaking, polyphyllins are glycosylated of triterpenoids. The complexity of polyphyllin structure impedes the chemical synthesis of these compounds for commercial use. Metabolic engineering is an alternative router for polyphyllin production. However, polyphyllin biosynthetic pathway has not been fully elucidated. In this manuscript, the authors utilized an integrated approach including phylogenetic, transcriptomic, metabolomics, gene co-expression, and biochemical analysis to characterize polyphyllin biosynthesis and identified two enzymes involved in producing polyphyllin. The research will be very interesting to scientists working on uncovering metabolic pathways of valuable plant-derived medicinal compounds. However, I think the manuscript needs substantial improvement before it can be accepted.

Major comments,

1. The first and second results should be consolidated. I don't think the Go annotation and KEGG analysis provide anything useful for the study rather than a pile of customized figures from the sequencing company.
2. To get a robust phylogenetic relationship, all the phylogenetic trees need to be performed using the maximum likelihood method with the bootstrap value shown in each branch. However, the Figure legend says it is Neighbor-Joining method, the MM says it is maximum likelihood. Which is the one the author used?
3. In Figure S10, how can the authors determine that the new peaks are indeed glycosylated at C-3? The standards used for co-chromatography are merely the non-glycosylated forms.

4. In figure 4d, there is a lack of details on how the trillin and pennogenin-3-O-glucoside standards are prepared

5. In Figure 2b, how is the control setup? I can see a peak corresponding to cycloartenol eluted around 19min in the control sample (which should be none). I am also a little confused about the result of PpOSC2. Looks like PpOSC2 produces a peak with the same retention time as cycloartenol. Did the author try to use the intensity of cycloartenol in the control as a background? If so, I think it is unacceptable because cycloartenol might naturally occur in *N.benthamiana*. The enzymatic assay with purified enzyme PpOSC1 and PpOSC2 might be necessary.

Minor comments

1. Line 669, the authors claimed that "steroidal sapogenin glucotransferase and steroidal glucotransferase are not the same genes by studying the enzyme kinetics of PpUGT73C1". I think the authors need to be careful with such a conclusion. The result only means PpUGT73C1 is better at catalyzing steroidal sapogenin. The possibility of another enzyme catalyzing both compounds can't be ruled out.

2. In the title, I suggest the authors emphasize the medicinal value of the plant instead of the giant genome.

3. Is it possible to include pictures of different organs of the plant used for transcriptomic and metabolomic analysis? It will make the paper more attractive to read.

4. Line 529, "infection" better to be replaced by "infiltration".

5. In Figure S10, it is better to include the chemical structures of the four compounds used for testing the enzyme activity.

6. In Figure 1, it is better to color the change of the structures after every enzymatic reaction.

To reviewer #1:

In this manuscript, the authors reported an effective method for predicting key genes of polyphyllins biosynthesis, and claimed their study provides a new idea for the study of gene cluster deficiency biosynthesis pathways in medicinal plants. But after reading this manuscript carefully, I find it difficult to evaluate the reliability of their workflow and the accuracy of the results, because a lot of the details of the analysis and results are not provided. For example, the software parameters and thresholds used in the analysis process (such as how to calculate the fpkm values using *deseq2*), and whether the important functional genes (*OSC*, *CYP*, and *UGT*) obtained by them have been submitted to GenBank, and where to find their sequences. I think only by providing these details can we continue to discuss the correctness of the manuscript, otherwise we can only reject it.

Response:

Thank you very much for the comments of the reviewers. In order to make our manuscript more clear, we have added some important details in the article, such as FPKM values determination (Please see page 7, line 142-146), some details of WGCAN (Please see page 8, line 185 and page 9, line 191-192 and 201-203). At the same time, we have added the key *OSC*, *CYP* and *UGT* genes sequences mentioned in the manuscript in the attachment for readers' reference, Please see supplementary files 2.

To reviewer #2:

- 1) It is better to modify the title of this manuscript, and exclude "none-clustered metabolic pathway".
- 2) Double check the manuscript, since there are a lot of mistakes. For examples, there is a mistake in the Figure 4d, it should be PpUGT73C1, instead of PpUGT73E5? In

addition, should it be clarified how PpUGT73C1 was selected for enzymic assay?

Response:

1. The title of the manuscript has been modified as follows according to the suggestions of the reviewer.
2. The manuscript has been checked again, some mistakes including Figure 4d have been corrected and marked with the revised mode.
3. All candidate UGT genes were tested for glucose glycosylation activity at c-3 position of Diosgenin, and UGT73C1 was the only gene with activity.

To reviewer #3:

Major comments

1. The first and second results should be consolidated. I don't think the Go annotation and KEGG analysis provide anything useful for the study rather than a pile of customized figures from the sequencing company.
2. To get a robust phylogenetic relationship, all the phylogenetic trees need to be performed using the maximum likelihood method with the bootstrap value shown in each branch. However, the Figure legend says it is Neighbor-Joining method, the MM says it is maximum likelihood. Which is the one the author used?
3. In Figure S10, how can the authors determine that the new peaks are indeed glycosylated at C-3? The standards used for co-chromatography are merely the non-glycosylated forms.
4. In figure 4d, there is a lack of details on how the trillin and pennogenin-3-O-glucoside

standards are prepared

5. In Figure 2b, how is the control setup? I can see a peak corresponding to cycloartenol eluted around 19min in the control sample (which should be none). I am also a little confused about the result of PpOSC2. Looks like PpOSC2 produces a peak with the same retention time as cycloartenol. Did the author try to use the intensity of cycloartenol in the control as a background? If so, I think it is unacceptable because cycloartenol might naturally occur in *N.benthamiana*. The enzymatic assay with purified enzyme PpOSC1 and PpOSC2 might be necessary.

Response:

1. According to the reviewer's comments, Go annotation and KEGG analysis were indeed not involved in the analysis of the metabolic pathways of polyphyllins, so this part of the results has been deleted.
2. All phylogenetic trees have been constructed according to Maximum Likelihood and bootstrap has been added to the phylogenetic tree.
3. As the research content of Figure S10 has been removed from the manuscript, there is no supplement for this part of the experiment.
4. Commercialized trillin and pennogenin was purchased from Chengdu Biopurify Phytochemicals Ltd. Because it is difficult to obtain the standard product of pennogenin-3-O-glucoside, we carried out nuclear magnetic detection on pennogenin and the obtained glycosylation product. By comparison, we found that the glycosylation product had a significant change at the C3 position, confirming glycosylation occurs at the C-3 position of pennogenin, please see Supplementary files 3.
5. In order to better verify the functions of *PpOSC1* and *PpOSC2*, yeast strains were used

to complete gene function verification. The results are consistent with those in *Nicotiana benthamiana*, and we found that *PpOSC1* gene had the function of cycloartenol synthase. These results have been updated in the manuscript.

Minor comments

1. Line 669, the authors claimed that “steroidal sapogenin glucotransferase and steroidal glucotransferase are not the same genes by studying the enzyme kinetics of PpUGT73C1”. I think the authors need to be careful with such a conclusion. The result only means PpUGT73C1 is better at catalyzing steroidal sapogenin. The possibility of another enzyme catalyzing both compounds can't be ruled out.
2. In the title, I suggest the authors emphasize the medicinal value of the plant instead of the giant genome.
3. Is it possible to include pictures of different organs of the plant used for transcriptomic and metabolomic analysis? It will make the paper more attractive to read.
4. Line 529, “infection” better to be replaced by “infiltration”.
5. In Figure S10, it is better to include the chemical structures of the four compounds used for testing the enzyme activity.
6. In Figure 1, it is better to color the change of the structures after every enzymatic reaction.

Response:

1. During the revision of the manuscript, we re-verified the gene function of UGTs and added new experimental data. After reviewers' suggestions, we believe that the previous “steroidal sapogenin glucotransferase and steroidal glucotransferase are not the same genes by studying the enzyme kinetics of PpUGT73C1” conclusion is not rigorous

enough. Since these results are not direct evidence for the interpretation of the synthesis pathway of polyphyllins, we chose to delete the content of this part.

2. The title of the article has been modified to " Multi-dimensional strategy speed dissection of biosynthetic pathway of bioactive polyphyllins".
3. According to the reviewer's comments, we have added a picture with annotations on each organs of the heavy building, please see Figure S1.
4. Done
5. This part of the content has been deleted from the manuscript.
6. Figure 1 has been redrawn as suggested by reviewer.

REVIEWERS' COMMENTS:

Reviewer #1 (Remarks to the Author):

1. I checked the website carefully, but I couldn't find Supplementary File 2 which contained the key OSC, CYP, and UGT gene sequences mentioned by the authors.
2. In Figure 1, the structures of the two molecules, Trillin and Pennogenin-3-O-Glu, are wrong.
3. In Line 466, I'm afraid there's something wrong with the unit of these Vmaxs, and usually, the unit of enzyme concentration should be μM , not mg.

Reviewer #2 (Remarks to the Author):

1. Authors indicated that all candidate UGT genes were tested for glucose glycosylation activity at c-3 position of Diosgenin, and UGT73C1 was the only gene with activity. I think this negative result should be put in the supplementary files, including protein express and enzymatic analysis.
2. Is floribundasaponin A in Figure 4 the same as pennogenin-3-O-Glu? They should be written consistently.
3. Can the pennogenin-3-O-Glu be detected in plants? If not, why do you think it is the product of PpUGT73C1 in *P. polyphylla* var. *yunnanensis*? This part of data should be shown or explained in the manuscript.

Reviewer #3 (Remarks to the Author):

The authors accepted most of my suggestions and the manuscript has been improved. However, a few questions remain to be addressed.

1. In addition to *N.benthamiana*, the authors used the yeast system to test the enzyme activity of PpOSC1. However, they didn't explain in the main text which yeast strain was used and whether it produces the substrate of OSC -- 2,3-oxidosqualene. Looks like the yeast strain information is in Fig. S3, but the reference Yuan et al. can't be located anywhere in the manuscript.
2. The authors said in the rebuttal letter that "we carried out nuclear magnetic detection on pennogenin and the obtained glycosylation product". However, this information as well as Supplementary files 3 is missing from the main text.
3. In the third panel of Figure 4d, the thick dashed line blocked the peak of floribundasaponin A.
4. The mass spectra of cycloartenol and PpOSC1 production in Figure 2b look identical. Please double-check.
5. In the legend of Figure 2b, " ...after being transferred to *N.benthamiana*"? What about Figure S8?

Dear reviewers,

Thanks for the suggestions on the article. Some mistakes and formatting problems in the manuscript have been modified according to the suggestions.

Reviewer 1:

1. I checked the website carefully, but I couldn't find Supplementary File 2 which contained the key OSC, CYP, and UGT gene sequences mentioned by the authors.
2. In Figure 1, the structures of the two molecules, Trillin and Pennogenin-3-O-Glu, are wrong.
3. In Line 466, I'm afraid there's something wrong with the unit of these Vmaxs, and usually, the unit of enzyme concentration should be uM, not mg.

Response:

1. Since only one supplementary file can be uploaded, the supplementary file 2 was not uploaded correctly. We have combined all supplementary materials into one file.
2. Figure 1 has been modified.
3. The units of Vmaxs have been converted to $\mu\text{M}/\text{min}$.

Reviewer 2

1. Authors indicated that all candidate UGT genes were tested for glucose glycosylation activity at c-3 position of Diosgenin, and UGT73C1 was the only gene with activity. I think this negative result should be put in the supplementary files, including protein express and enzymatic analysis.

2. Is floribundasaponin A in Figure 4 the same as pennogenin-3-O-Glu? They should be written consistently.
3. Can the pennogenin-3-O-Glu be detected in plants? If not, why do you think it is the product of PpUGT73C1 in *P. polyphylla* var. *yunnanensis*? This part of data should be shown or explained in the manuscript.

Response:

1. According to the requirements of reviewers, we have supplemented these experimental results. We screened 26 glycosyltransferase genes (Fig. 4a, marked with asterisks) that may be involved in the synthesis of polyphyllins through WGCNA analysis. Except for the functionally characterized PpUGT73CR1 and four PpUGTs (F01_transcript/43033, 12407, 23323, 48334) that were not cloned from cDNA, the protein expression and enzymatic analysis results of the remaining 21 UGT genes were added as Figure S9 and S10.
2. Pennogenin-3-o-glu is the same compound as floribundasaponin A. We have replaced all pennogenin-3-O-Glu in the manuscript and pictures with floribundasaponin A.
3. Floribundasaponin A is present in plants, and it has been reported that floribundasaponin A can be found in *Dioscorjgi Floribunda* (Shashib Mahato et al., 1981, 20:8, 1943-1946) and *Paris polyphylla* var. *chinensis* (Yoshihiro MIMAK et al., Natural Product Letters, 2000, 14:5, 357-364).

Reviewer 3

1. In addition to *N.benthamiana*, the authors used the yeast system to test the enzyme

activity of PpOSC1. However, they didn't explain in the main text which yeast strain was used and whether it produces the substrate of OSC -- 2,3-oxidosqualene. Looks like the yeast strain information is in Fig. S3, but the reference Yuan et al. can't be located anywhere in the manuscript.

2. The authors said in the rebuttal letter that "we carried out nuclear magnetic detection on pennogenin and the obtained glycosylation product". However, this information as well as Supplementary files 3 is missing from the main text.

3. In the third panel of Figure 4d, the thick dashed line blocked the peak of floribundasaponin A.

4. The mass spectra of cycloartenol and PpOSC1 production in Figure 2b look identical. Please double-check.

5. In the legend of Figure 2b, "...after being transferred to N.benthamiana"? What about Figure S8?

Response:

1. Details of yeast strains have been added to the text, and relevant references have been cited in the supplement file 1.

2. Since only one supplementary file can be uploaded, the supplementary file 3 was not uploaded correctly. We have combined all supplementary materials into one file.

3. In Figure 4d, we have replaced the thick dashed line with the name of compound.

4. We have carefully checked the two mass spectra and we have determined that these two mass spectra are from cycloartenol and PpOSC1 respectively.

5. N.benthamiana in Figure 2B is a clerical error, we have changed it to yeast strain SQ-

pPOSC1.